# Chemical Sensors for Farm-to-Table Monitoring of Fruit Quality

**DOI:** 10.3390/s21051634

**Published:** 2021-02-26

**Authors:** Denise Wilson

**Affiliations:** Department of Electrical and Computer Engineering, University of Washington, Seattle, WA 98195-2500, USA; denisew@uw.edu; Tel.: +1-206-221-5238

**Keywords:** farm-to-table, community-supported agriculture, sustainable agriculture, pH sensors, Brix sensors, sugar content, ethylene sensors

## Abstract

Farm-to-table operations produce, transport, and deliver produce to consumers in very different ways than conventional, corporate-scale agriculture operations. As a result, the time it takes to get a freshly picked fruit to the consumer is relatively short and the expectations of the consumer for freshness and quality are high. Since many of these operations involve small farms and small businesses, resources to deploy sensors and instruments for monitoring quality are scarce compared to larger operations. Within stringent power, cost, and size constraints, this article analyzes chemical sensor technologies suitable for monitoring fruit quality from the point of harvest to consumption in farm-to-table operations. Approaches to measuring sweetness (sugar content), acidity (pH), and ethylene gas are emphasized. Not surprisingly, many instruments developed for laboratory use or larger-scale operations are not suitable for farm-to-table operations. However, there are many opportunities still available to adapt pH, sugar, and ethylene sensing to the unique needs of localized farm-to-table operations that can help these operations survive and expand well into the future.

## 1. Introduction

At its core, farm-to-table is a social movement organized to provide economic benefits to local communities, reduce the climate and environmental impacts of growing food for human consumption, and improve the nutritional value and flavor of food at the point of consumption. Farm-to-table also goes by other similar names including farm-to-fork, farm-to-school, locally-sourced, and farm-to-cafeteria. While not a regulated term, farm-to-table is characterized by food that reaches the point of consumption directly from a farm rather than going through a store, distributor, storage, or other stop along the way. Food grown for human consumption can travel through the shortened, local supply chain characteristic of farm-to-table practices through community-supported agriculture (CSA), a farmer’s market, or other direct sales relationship that allows individual consumers, restaurants, or other food service businesses to obtain food directly from known, reliable, and local sources.

Farm-to-table significantly reduces the carbon footprint involved in food transport and distribution. Nearly half of all the fruit sold in the U.S. is imported and even produce grown in North America travels an average of 2,000 km from where it is farmed to where it is sold [1]. Despite the long distances that fruit and vegetables travel to reach the consumer, transportation of food accounts for only 11% of the greenhouse gas emissions incurred by food production [2]. Nevertheless, farm-to-table operations require food to be transported over much smaller distances than food produced by corporate agriculture or imported from other countries. Shorter distances require less diesel and other fossil fuel consumption and reduce the carbon footprint of the fruit production life cycle. As a percentage of overall greenhouse gas emissions, these reductions are even larger for fruit and vegetables per kg than for meat production because fruits require far less land use change and do not incur the significant methane emissions involved in raising cattle and other farm animals [3]. 

As or more important than greenhouse gas emissions incurred when fruit must travel long distances to reach the point of consumption are the nutrients lost between harvest and consumption. Fruits and vegetables typically contain over 90% water and once they are harvested, respiration rates increase, moisture decreases, and fruit quality degrades rapidly [4]. While refrigeration can slow down respiration processes and allow for longer shelf lives, nutrients inevitably degrade with time after harvest [4]. Of all nutrients lost with storage or transport of fresh fruit and vegetables, post-harvest decreases in vitamin C are the most dramatic, but riboflavin, folic acid, pantothenic acid, and biotin are also highly sensitive to post-harvest conditions and decline with increased temperatures and storage times [5]. However, in lemons and grapefruit, very little vitamin C is lost during storage and in reasonable storage conditions, losses of vitamin C for other citrus fruits are quite small. Unlike vitamin C, precursors to vitamin A (e.g., carotene) are stable and suffer little during storage [5]. In contrast, polyphenolics (e.g., flavonoids) decrease significantly with the storage of fresh produce [6]. Thus, while conventional wisdom may promote the idea that a freshly picked fruit is always more nutritious than one stored, evidence suggests that nutrient degradation after harvest is much more of a mixed bag. Nevertheless, combined with small farm practices, both fruits and vegetables supplied by farm-to-table operations are likely to offer greater nutritional value and flavor than produce generated by corporate agriculture operations.

Farm-to-table operations also benefit local economies and small farms. Between 1948 and 2015 in the United States, four million farms disappeared even as farm output doubled [7]. While this statistic alone underscores the crisis that small farms face in modern agriculture, the rise in demand for fresher and better tasting food among consumers has both helped and hindered the fight that small farms face to survive. In order to be successful in farm-to-table, small farms must diversify the fruits, vegetables, and other products that they produce, devise strategies to supply products such as cold storage vegetables and meat during the winter months, and invest in multiple supply chains ranging from direct-to-consumer farmers markets to restaurant supported agriculture [8]. Ensuring that consistently high quality of produce are delivered to a myriad of customers will continue to enable the farm-to-table movement to expand. 

As with corporate agriculture, the judicious use of sensor technologies in the monitoring, harvest, sorting, and transport of farm-to-table products can support greater yields, reduce waste, and improve profit margins. However, what needs to be sensed and when can be drastically different than the longer, more complex and more controlled food supply chains inherent to corporate agriculture. This article takes a closer look at the farm-to-table supply chain and opportunities for sensors to support the continued optimization and expansion of this important social movement. 

In the context of farm-to-table operations, the chemical composition of fruits and vegetables is clearly a major contributor to the exceptional taste, flavor, and overall quality expected of localized food production operations. However, the chemical composition of fruits is complex and varies widely among different fruit types and cultivars. For fruits, how sweet, how acidic, and how ripe a fruit is as well as how these qualities change over time are very important to maximizing the value of what is delivered to the consumer, whether at the restaurant or at home. Therefore, this review of chemical sensor technologies focuses on these three parameters (sweetness/sugar content, acidity/pH, and ripeness/ethylene emissions) not because they are all that matters in determining fruit quality but because they are common to all fruits in influencing overall quality. 

## 2. Chemistry of Interest in Farm-to-Table Operations

In farm-to-table operations, the quality of individual fruit is far more important than it is in higher volume and canning/preserving operations. The chemical composition of the fruit is only part of what is used to determine both perceived and actual quality. Other parameters such as color, shape, texture, and homogeneity also influence the perception of quality. Almost everyone loves a lush red, symmetric tomato, but some attributes such as color also indicate nutritional quality. For instance, the red color in the traditional tomato is highly correlated to the amount of lycopene it contains and the many nutritional benefits that lycopene offers to human health [9]. Furthermore, color has long been used as an indicator of how mature a fruit is and plays a role in determining the optimal point of harvest [10]. In combination, color and firmness are also used together to not only track maturation of the fruit prior to harvest but to predict and monitor post-harvest degradation [11]. Ironically, many of these visible indicators are often used by the consumer as the sole indicators of quality while the underlying chemical composition which determines these indicators and their change over time is what matters much more in terms of flavor and nutritional value. Chemical sensors offer the opportunity to make the invisible chemistry of plants, produce, and food visible and data collected from these sensors can complement visual indicators of fruit quality. 

### 2.1. Sugar Content

Sugar plays an important role in what makes a fruit taste good and is a critical parameter for monitoring and determining the quality of fruit as it travels from farm to table. Many fruits are mostly water but what remains after water is taken out of the equation is dominated by different types of sugar whose concentrations evolve during the ripening process. For example, while a tomato is between 90% and 95% water, 50% of what remains is sugar and is represented by glucose, fructose, and sucrose. Sugar is a major component of tomato flavor both in terms of the total amount of sugar in the tomato and in the amount relative to the acid also present in the tomato. During ripening, sugar content increases during the ripening process and by the time a tomato is mature, glucose and fructose contribute approximately equal amounts to the total sugar content while sucrose is a more minor contributor [12]. Furthermore, degradation of tomato acids during maturation and ripening leads to the accumulation of sugars in the fruit. Thus, not only are sugar and acid content largely responsible for fruit sweetness and sourness but the balance between them is an indication of ripening stage. While total sugar content does not tell the full story regarding the quality and taste of fruit, it is nevertheless an important parameter to measure both in fruit and fruit juice and many instruments for measuring it are commercially available. Total sugar content is often expressed in units of degrees Brix. One degree Brix is equal to one gram of sucrose in 100 g of solution or 1% sucrose by mass. Multiplying degrees Brix by 10 gives the amount of sucrose in solution by volume. One degree Brix is therefore equal to 10 g of sucrose per liter of solution. In the case of fruit, solution refers to either a fruit juice or a mash made by processing the intact fruit until it approximates a solution. Degrees Brix (°Brix) have been extensively characterized according to quality of fruit and the change in °Brix over time can be used as a valuable indicator of the maturation and over-ripening of fruit during transport and storage. Estimates of fruit quality according to sugar content often use the broadly accepted Brix chart formulated by agricultural engineer Dr. Carey Reams. A representative sampling of fruits from the Brix chart is provided in Table 1 [13].

While Brix values are important, they do not tell the whole story when it comes to the flavor and quality of fruit. Brix measurements assume that the dissolved solids that are being measured consist entirely of sucrose. While in most fruits, sugars are the majority of soluble solids, other soluble solids include carbohydrates, organic and amino acids, proteins, fats, and minerals. The presence of these other soluble solids can lead to over-estimates of the amount of sugar in fruits and fruit juices. For example, in some mangoes, total soluble solids content (TSS) at the point of harvest are approximately 8.7% while the sugar content is only 6.1% [14]. Similarly, the TSS of certain pears can be as high as 23.0% while the total sugar content is only 10.4% [15]. Furthermore, fruits contain sugars other than sucrose which lead to additional errors in Brix measurements. For instance, tomatoes contain 1.1 g of glucose, 1.14 g of fructose, and 0 g of sucrose, while peaches contain 1.1 g of glucose, 1.3 g of fructose and 5.6 g of sucrose per 100 g of ripe fruit [16]. Further, in the big picture of fruit flavor and perceived quality, an understanding of both acidity and sugar content is necessary to more fully understand the taste of fruit and its role in overall perceived quality. In farm-to-table operations where the value of flavor is at a premium, measuring both acidity and sugar content is critical to accurately gauging flavor. 

### 2.2. PH and Acidity

Acidity in fruit can be measured either by sensing pH in the juice of the fruit or by measuring titratable acidity. Titratable acidity measurements typically require adding a known basic solution to the fruit juice until the solution transitions from an acid to neutral. The amount of base added to neutralize the solution is then a measure of how much acid was present in the original juice. The titration process takes time and is best suited to lab or benchtop situations rather than in-field or other in situ measurements. In contrast, pH is much easier to measure and a wide range of sensing technologies and instrument designs are available to measure pH in the field. pH stands for power of hydrogen and represents the concentration of hydrogen ions in solution. pH is measured on a scale between 1 and 14, with lower numbers below 6 representing acids and numbers above 9 representing strong bases. Fruits are primarily acidic (Table 2), but some such as cantaloupe and honeydew melon are considered low-acid food, with pH values over 6 but below neutral pH (i.e., 7 pH units). 

In and of itself, pH is important as an indicator of food safety in canned and preserved fruits. For example, among the over 12 million tons of tomatoes grown every year in the United States for use in processed products [17], pH plays a critical role in yield and quality. Among these tomatoes, the pH must remain low enough to ensure food safety but also high enough to maintain flavor. Despite the fact that tomatoes are not a low-acid food, they must nevertheless be maintained at a pH between 4.2 and 4.3 in canned tomatoes, tomato paste, catsup, and similar products [18]. A maximum safe level of 4.4 with an optimal target pH of 4.2–4.3 has been suggested to prevent *Clostridium botulinum* (botulism) spores from growing and producing toxin [19].

However, of much greater relevance to fruit in farm-to-table operations is a combination of pH, sugar content, and flavor volatiles, which together become the taste and flavor of the fruit [21] and also play an important role in dictating the maturation, ripening, and over-ripening of fruit pre- and post-harvest. Both the types of acid in and the nature of pH changes in fruit can vary widely from one type and cultivar to the next. For example, citrates in peppers and tomatoes both increase during maturation and ripening, but in tomatoes, malates increase until just before full maturity, before decreasing until the tomato reaches optimal ripening conditions. In peppers, however, the trend is opposite. Malates decrease and then increase again to full ripening. Simultaneously, sucrose sugar content decreases during ripening for both tomatoes and peppers while fructose and glucose sugars increase [22]. Furthermore, the dominant organic acid in a fruit varies according to species with malic acid a major player in such fruit as apples [23] and citric acids important among citrus fruits [24]. Even among different cultivars of the same fruit (e.g., pears [25], bananas [26], peach [27]), both total acidity and the mixture of organic acids that contribute to acidity are distinct. 

Thus, while sensing pH is directly useful for predicting the food safety of canned or preserved products, it is not particularly useful as a sole measurement for fresh fruit. When used in combination with the measurement of sugar content, however, sensing pH can provide a much more relevant (although still not complete) picture of fruit quality. Accessible, low-cost pH sensing technology can therefore play a valuable role in farm-to-table operations. 

### 2.3. Ethylene Emission and Absorption

In terms of ripening, there are two major types of fruit. Those that continue to ripen after harvest (e.g., apples, tomatoes) are called climacteric while those that do not ripen after harvest (e.g., strawberries, grapes) are termed non-climacteric. The ripening process in climacteric fruit is controlled by the production of ethylene gas, a phytohormone (i.e., plant hormone) that acts as a chemical messenger to regulate the cellular activities involved in ripening. In the climacteric fruit, ethylene production increases during maturation, rising sharply right before optimal ripening and then decreasing dramatically as the fruit ages, over-ripens, and degrades. In contrast, while non-climacteric fruits also produce some ethylene gas, they do not experience the sharp increases in ethylene production that characterize the ripening of climacteric fruit nor the dramatic decreases that occur during over-ripening (Figure 1). 

And, as is the case with pH and sugar content, the amount of ethylene produced and the sensitivity of fruits to ethylene can vary widely with species and cultivar (Table 3). Recent research has shown that the distinction between climacteric and non-climacteric fruit is blurred, as many climacteric fruits (e.g., pears) now exhibit some non-climacteric behaviors or include non-climacteric cultivars and the opposite is true for some non-climacteric fruits (e.g., strawberries) [28].

In conventional agriculture, ethylene emissions by fruits are often inhibited by the introduction of 1-methylcyclopropene (1-MCP) to delay ripening or are externally controlled with additional ethylene gas to speed up ripening after transport and prior to sale [32]. While these steps are largely unneeded in farm-to-table operations, natural ethylene emissions from fruits and vegetables can still cause deterioration and over-ripening, particularly when moderate to high ethylene emitters are packaged or stored with ethylene-sensitive produce. For example, broccoli and apples are both in season in the fall months, while the former is ethylene sensitive and the latter is an ethylene producer [33]. Packaging both in the same farm box for delivery to residential households or storing both in the same place at a restaurant can lead to premature yellowing of broccoli, loss of nutrients, and reductions in flavor. Thus, it is important to know both the ambient ethylene gas concentration among fruits during transport and storage as well as the type of fruits that are in close proximity to one another. Tracking the rate of change of ethylene production among climacteric fruits at the point of harvest is also very useful in predicting ethylene exposure during post-harvest operations. If transport and storage times and conditions (e.g., temperature and humidity) are known, these gradients in ethylene production can be used to harvest fruit at such a time that optimal freshness is reached at the point of consumption. In farm-to-table operations, customer expectations for fruit freshness are often higher than for fruits which traverse a longer supply chain and which often undergo artificial manipulation of the ripening process en route. Because of these increased expectations, monitoring ambient ethylene gas concentration among farm-to-table fruit can be as or more important in establishing, tracking, and predicting fruit quality as it is for corporate fruit production. 

Ethylene sensors can be useful for reducing the detrimental impacts of ethylene gas in farm-to-table operations in three general ways: (a) monitoring ethylene gas pre-harvest to determine optimal ripening; (b) characterizing amount and temporal variations in ethylene emissions from mixtures of various fruits and vegetables in experimental research settings to provide guidance to small farms on what and how much can be packaged together; or (c) deploying portable ethylene sensors to monitor fruits and vegetables whether or not they are transported or stored together. The former approach allows for established benchtop or laboratory methods of ethylene sensing to be viable while the latter approach requires low-cost, low-power, portable, and small sensors. 

## 3. Sensing Technologies for Sweetness (Sugar Content)

Sugar content in fruit is typically measured in degrees Brix, where one degree represents one gram of sucrose (sugar) in 100 g of solution. Most sensors used to measure sugar content assume that most or all of the sugars present in a sample are sucrose. When this is not the case, small errors introduced by other sugars (e.g., fructose, glucose) are usually expected and tolerated. Many methods also sense sugar content indirectly by measuring the total dissolved solids in a liquid sample. These indirect methods are vulnerable to interference from other dissolved solids (e.g., proteins, fats, minerals), but also enable simple, low-power, and inexpensive instruments. Therefore, they remain popular in estimating the sugar content not only of fruit and fruit juice but also of wine, honey, carbonated beverages, maple syrup, and similar products. Indirect methods of determining degrees Brix (i.e., sucrose content) include measuring the specific gravity of a solution using a hydrometer or oscillating U-shaped tube meter, or alternatively, measuring refractive index (RI) using a refractometer. Optical sensor modalities such as surface plasmon resonance compete with traditional refractometer instruments to offer very high resolutions in determining RI. However, these indirect methods share the common problem of requiring that juice be extracted from the fruit or that the fruit be processed into a mash that is sufficiently liquid and homogenous for a reliable measurement. This process is generally destructive to the fruit itself and particularly so for firm fruits which are difficult to mash. Microextraction methods and miniaturized measurement instruments have the potential to draw sufficiently small sample volumes to enable non-destructive evaluation of fruit sweetness using these measurement methods. At the present time, however, such non-destructive approaches to sampling fruit and measuring degrees Brix are not commercially available. However, sugar sensing methods that capitalize on the absorption of infrared (IR) light and the speed of sound inside a fruit can be performed non-destructively. IR methods offer the added benefit of a more direct measure of sucrose content that is more resistant to interference from related compounds or other dissolved solids as compared to indirect sensing methods that use specific gravity or RI to infer °Brix. 

### 3.1. Hydrometers and U-Shaped Tube Meters

Both hydrometers and oscillating U-shaped tube meters estimate sugar content by measuring specific gravity. When applied to liquids, specific gravity is the ratio of the density of a liquid relative to that of water when it is in its most dense state (at 4 °C). The specific gravity of typical fruit juices varies between 1.02 and 1.05 [34] and is linearly related to degrees Brix within this range [35]:(1)°Brix ~248*Specific Gravity−248

The hydrometer (Figure 2a) uses buoyancy to estimate specific gravity by inserting the instrument into a liquid of unknown specific gravity until it floats. The greater the density and specific gravity of a fruit juice, the denser the solution is and the higher the hydrometer bulb will float. Hydrometers are typically customized to the range of specific gravities expected in the measurement process and can cost anywhere from $10 for home winemaking and beer brewing [36] to hundreds of dollars for instruments that measure specific gravity in 0.001 increments [37]. 

An oscillating U-shaped tube (Figure 2b) can also be used to measure specific gravity. A liquid sample (e.g fruit juice or fruit mash) fills a U-shaped tube and the tube is electrically excited with a piezoelectric actuator which stimulates the liquid to oscillate in the tube. The period of oscillation is related to the density and specific gravity of the liquid and is sensed by using two optical sensors on either side of the U-shaped tube. Density and specific gravity of liquids can be measured with precision and resolution as high as 10^−6^ using the oscillating U-shaped tube but these instruments often cost thousands of dollars and are prohibitively expensive and too large [38] for farm-to-table applications. 

Using hydrometers or oscillating U-shaped tube meters to measure sugar content presumes that the only contributor to specific gravity changes is sucrose. Thus, regardless of the resolution and precision of the specific gravity measurement itself, this approach to measuring sugar content is vulnerable to inaccuracies introduced by the presence of other molecules or solids that influence specific gravity as well as by the type and composition of sugar present in a sample. 

### 3.2. Refractive Index Sensors

As sugar (e.g., sucrose) concentration in a liquid solution (e.g., fruit juice) increases, light entering into the solution travels more slowly and bends (refracts) more. This behavior is expressed as an increase in the refractive index (RI) of the solution. Except for some small curvature at high concentrations, the RI of sucrose in solution increases linearly with increasing sucrose concentration at a given wavelength and temperature. For example, at a range of °Brix relevant to fruit (corresponding to 0 to 15% dissolved sugar) and for a wavelength of 589 nm at a temperature of 20 °C, degrees Brix can be estimated according to the following expression [39]:(2)°Brix ~667*RI−889

Temperature has a significant influence on RI—on the order of a 10^−4^ decrease in RI units per one °C increase in temperature for water-based solutions [40]. 

A traditional RI sensor, called a refractometer, directs light through a liquid sample to a prism that is in contact with a sample (e.g., fruit juice) along a wide range of incident angles (Figure 3). When the incoming light reaches the interface between the prism and the sample at an angle that is less than the critical angle, some light is refracted and some light is reflected onto the light sensor array. At angles greater than the critical angle, all light is reflected onto the array. If the human eye is used instead of an electronic array of light sensors, the refractometer can be operated at zero power, as is the case with a handheld or analog refractometer. Abbe refractometers use two prisms, on-board temperature sensors, and circulating water to control temperature for high-precision measurements of refractive index. 

Analog refractometers are quite inexpensive, offer precision on the order of 10^−4^ RI units (0.1 °Brix), and are available commercially for less than $300; their portability, zero power consumption, and low cost make them well suited to monitoring fruit at the point of harvest. Digital refractometers cost more (on the order of $500), consume power, are less vulnerable to human error, and offer similar precision (0.1 °Brix) as analog units. Abbe refractometers offer as good or better resolution to digital and analog refractometers and greater accuracy in response to ambient temperature fluctuations but are designed for laboratory use and are prohibitively expensive for most farm-to-table applications. 

A number of alternative optical sensor technologies for measuring RI (and by extension sugar content in fruit juice) have been demonstrated in the literature. For example, saurface plasmon resonance (SPR) is a physical process by which energy from incoming light is coupled to energy associated with free electrons in a noble metal such as gold. At a fixed wavelength, only light incident upon the metal at a particular angle is optimally absorbed by the free electrons and converted to surface plasmons. Surface plasmons are delocalized oscillations of electrons that occur at the interface of two materials whose dielectric functions have real components which are opposite sign (i.e., one positive and one negative) such as is the case between a noble metal and air. The angle at which optimal or maximum absorption of the light energy occurs is called the point of surface plasmon resonance and is dependent on the RI of the material in contact with the noble metal. Alternatively, if the angle of incident light is held constant, SPR can be measured by finding the wavelength of light at which the maximum amount of light energy is absorbed by surface plasmons. In either case, SPR is highly sensitive to refractive index, delivering resolutions as high as 1 × 10^−7^ to 2 × 10^−7^ RI units at wavelengths between 700 and 900 nm [41]. Similar resolutions are possible with fused silica microsphere resonators where light travels in the form of whispering gallery modes (WGM) along the curved surface of the spheres. Similar to fixed angle SPR, the wavelength of these resonant WGM resonators is very sensitive to the RI of the surrounding medium, providing RI resolution as low as 1 × 10^−7^ RI units [42]. Recent advances in microsphere design have enabled the discrimination of temperature changes from RI changes in the liquid sample, thus providing much needed improvements in accuracy when these RI sensors are used in field environments where ambient temperatures can vary widely [43]. Other optical structures are also sensitive to RI changes but deliver lower RI resolution. Those based on photonic crystal microcavities can detect RI at 10^−3^ RI units of resolution in liquids [44] and 10^−4^ RI unit resolutions in gases [45]. Photonic crystal fibers containing long period gratings have been demonstrated with RI resolutions of 8.5 × 10^−6^ RI units [46]. 

### 3.3. Alternative Approaches

Another indirect approach to measuring sugar content in solution involves the use of capacitive sensors. Capacitance depends on the permittivity of the dielectric between the two measurement plates (i.e., electrodes) of a capacitor. Permittivity changes with sugar content in solution, and the permittivity of both water and sucrose varies with frequency with larger differences between sugar and water evident at lower frequencies (below 1 MHz). These differences can be exploited to generate characteristic frequency vs. voltage curves for sucrose solutions that provide estimates of the amount of sucrose in solution. This method has been successfully demonstrated for detecting sucrose in solution at concentrations between 10% (100 g/L) and 50% (500 g/L) [47] and between 9.0 °Brix and 10.9 °Brix in orange fruit [48]. As an indirect measure of sugar concentration, capacitive-based sugar sensors are as vulnerable as all other indirect sensing methods to interference from other solids and different compositions of sugar in solution. 

Ultrasound provides yet another alternative to measuring sugar content, density, and refractive index in fruit and fruit juices. Sound travels more slowly in materials that are denser. Increasing sugar content is linked to increasing fruit density and slower speed of sound through the fruit. This property has been used to measure sugar content in fruit juice across a range of 0–40 °Brix within an error of 0.2% for juices that are dominated by a single sugar and 0.5% for juices that contain multiple types of sugar [49]. Ultrasonic measurements have also been used to identify sugar content in intact fruits or fruit samples including mangos, tomatoes, melons, and plums [50]. As with many other methods to measuring sugar, ultrasound is prone to interference from other dissolved solids and also demonstrates limited accuracy compared to RI-based sensors. 

### 3.4. Summary of non-Spectroscopic Technologies 

Multiple sensor technologies are available to measure sugar content in fruits and fruit juices including those based on specific gravity (hydrometers, oscillating U-shaped tube meters), RI (analog, digital, Abbe, SPR, microsphere), capacitance, and the speed of sound. Several of these methods provide very high accuracy and resolution across a range of °Brix relevant to fruits and fruit juices (Table 4). However, common to all of these approaches is an underlying assumption that changes in the output parameter (e.g., specific gravity, RI) are dominated by variations in sucrose content in the fruit or fruit juice. The presence of other sugars, particularly fructose and glucose, introduce interference and error to these sensing approaches. Further, while these sensors and instruments can be used non-destructively in the evaluation of fruit juice, the same is not the case for intact fruit. In order to estimate sugar content, the fruit must be converted to a mash which inherently destroys the fruit in the process. While this may work for sampling of sweetness in batches of fruit, monitoring of the sweetness or sugar content of individual fruit requires a non-destructive sensing or sampling technique. Of the major sensing technologies available to evaluate sugar content non-destructively, light spectroscopy is one of the only viable options. Because of the absorption characteristics of sugar, light in the near-infrared region of the electromagnetic spectrum provides the best choice of wavelengths for analyzing sugar content in fruit. Near-infrared (NIR) spectroscopy methods are discussed next.

### 3.5. Near-Infrared (NIR) Spectroscopy

Limited by laws of quantum mechanics, the absorption of near-infrared light by solid materials is limited to overtone and combination vibrations in a molecule. Because these vibrations are limited, molar absorptivity in the NIR region of the electromagnetic spectrum is small and provides only low sensitivity measurements. However, the absorption of common sugars (sucrose, fructose, glucose) is distinct in the NIR region and exhibits subtle variations around a wavelength of 960 nm that make it possible to distinguish among them [57]. Further, the fact that NIR penetrates solid materials to larger depths than other forms of IR spectroscopy makes it possible to analyze solid materials such as fruit without a lot of sample preparation. Further, among the shorter wavelengths of the NIR spectrum, conventional silicon photodetectors may be used to detect transmitted or reflected NIR light, thus opening the door for affordable and portable NIR instruments that can take advantage of the low cost of silicon relative to other semiconductor photodetectors. Several handheld NIR spectrometers are presently available commercially although their performance and usefulness has yielded mixed reviews [58] and these instruments can easily cost into the thousands of dollars. 

The advantages of NIR approaches to detecting sugar content have led to numerous studies of fruit using NIR spectroscopy. The response of fruits to NIR can be evaluated in multiple ways. NIR absorption spectroscopy studies how much light is absorbed while transmittance (Figure 4a) evaluates how much NIR is transmitted through a liquid or solid. Reflectance (Figure 4b) measures the light that is reflected directly from the surface of a fruit while diffuse reflectance (Figure 4c) measures light reflected at multiple angles from the surface. Interactance (Figure 4d) measures light that has scattered underneath the surface and returned to the fruit surface and is a better representation of what is going on internally. 

Unlike other sensor technologies used to estimate the sugar content of fruit and fruit juice, results of NIR studies often report their results in terms of R, R^2^, SEP and RMSEP (Table 5). 

R is the correlation coefficient. R^2^ is the coefficient of determination which indicates how much variation in the NIR method is attributed to actual variation in the parameter being measured as established by calibration with an accepted reference such as Fourier transform infrared spectroscopy (FTIR). For example, an R^2^ value of 0.98 indicates that 2% of the variation in a set of NIR measurements comes from error while the remaining variation comes from actual changes in the parameter being measured (e.g., total sugar content). RMSEP is the root mean square error of prediction and represents the average uncertainty that can be expected when evaluating samples outside of the calibration set (i.e., new samples). For detecting total sugar content, the accuracy of any future prediction of sugar content is then within +/− twice the RMSEP. SEP, on the other hand, measures the precision of the prediction or the amount that a future concentration will vary when making the same measurement multiple times (i.e., repeated measurements). Using these metrics, a wide range of fruits and fruit juices have been evaluated, often with R^2^ values above 0.90 and prediction errors of less than 1 when measuring total sugars, soluble solids content (SSC) as a proxy for total sugars, and °Brix (Table 5). 

## 4. Sensing Technologies for Acidity (pH)

pH sensors can be broken down into a number of categories: (a) zero power, low-resolution papers and strips; (b) glass electrodes and other electrochemical sensors; (c) higher-cost optical instruments; and (d) other electrical and electromechanical sensor technologies which have the potential to be commercialized as low-cost, portable alternatives to existing pH sensors on the market. At the time of this writing, the vast majority of commercialized pH sensors belong to categories (a) and (b). As with sugar content (sweetness) sensors, pH sensing technologies require that the fruit sample be liquid, either as fruit juice or as a mash of the intact fruit. Most existing pH sensors require destruction of the fruit to prepare a suitable liquid sample to determine pH. Microextraction of solid samples from firm fruits (e.g., apples) or the juice internal to softer fruit (e.g., tomatoes) would enable pH to be detected non-destructively, but techniques to do so are limited and not widely commercially available. 

### 4.1. Traditional pH Sensors (Papers and Strips)

The oldest and most inexpensive pH sensors are the litmus indicators, the first of which were extracted from lichens that turn red when exposed to an acid and blue when exposed to a base. Commercial indicators that are relevant to monitoring fruit acidity include methyl red, methyl orange, and bromocresol green which offer color transitions within a pH unit of 5.1, 3.7, and 4.7 pH units, respectively [72]. These indicators offer coarse pH measurements at resolutions of 1.0 pH units or greater across a narrow range of pH. While 1 pH unit may be sufficient to differentiate an unripe fruit from a ripe one, greater resolution is necessary to make the kinds of distinctions necessary to monitor acidity relevant to fruit flavor and overall fruit quality at the point of harvest and beyond. 

### 4.2. Glass Electrodes for Measuring pH

A widespread, higher resolution, commercial alternative to sense pH is the glass electrode (Figure 5a). The glass electrode works by using an electrochemical cell to measure the concentration of hydrogen ions in solution. Hydrogen ions in solution interact with a solid material, called a working electrode, to either extract electrons or impart electrons to the electrode. The number of electrons that are exchanged between the solution and the working electrode depends on the concentration of hydrogen ions in the solution. pH can be calculated from the hydrogen ion concentration using the following relationship:(3)$$pH=-\log_{10}\left(\left[H^{+}\right]\right)$$

Another electrode in the system, the reference electrode, provides a stable reference point to the electrochemical cell in a manner that is similar to a signal ground in an electrical circuit. A stable equilibrium is maintained between the reference electrode and the electrolyte surrounding it by design. In this way, the resulting potential difference between working and reference electrode provides a reliable and stable indicator of the accumulation or extraction of electrons by hydrogen ions at the working electrode (i.e., pH). 

A typical electrochemical pH electrode measurement system uses an Ag/AgCl (silver/silver chloride) reference electrode in a constant pH solution as the reference electrode and reference solution, respectively. This establishes the reference potential. A glass electrode coated with a hydrated gel then serves as the working electrode, and the potential across the two electrodes (reference and working) is given by the Nernst equation:(4)$$V=V_{o}-\;\frac{2.3RT}{nF}\left(pH\right)$$
where *R* and *F* are constants, *n* is constant for a given type of electrode or pH sensor, *T* is temperature in degrees Kelvin, *V*_o_ is a constant that depends on the reference electrode, and *pH* is the unknown pH in the liquid sample. 

Glass electrode pH measurement systems are commercially available from a range of vendors including Hach, Fisher Scientific, Cole-Parmer, Hanna Instruments, and Mettler Toledo and range in cost between $100 and $500 USD. Glass electrodes are fragile, relatively large, and require frequent recalibration which limit their use for farm-to-table operations. Regardless, among the possible approaches to measuring pH, the glass electrode has been the most successful commercially and is likely to be a candidate in any agricultural application that requires accurate and reliable measurements of pH. 

### 4.3. Other Electrochemical Approaches to pH Measurement

Over several decades of research, various possibilities for miniaturizing electrochemical sensors have been investigated as compact, low-cost, robust alternatives to the glass electrode [73]. Miniaturizing pH sensors is a critical step for enabling these sensors to be used in many of the harvesting, transport, and storage processes in farm-to-table operations. The most promising option for miniaturizing the glass electrode is the ISFET (ion-sensitive field effect transistor) shown schematically in Figure 5b. In a pH-sensitive ISFET, the gate of the transistor is removed and the underlying insulator layer interacts with the liquid sample, either accumulating additional electrons or losing electrons in doing so. Much like the working electrode in the conventional glass electrode, the resulting charge on the insulator layer is proportional to the concentration of hydrogen ions in and pH of the solution in a manner that follows the Nernst equation (Equation (4)). Similar to the operation of a standard MOSFET in computer chips, the charge on the insulator layer then modulates the current travelling along the channel of the transistor (between drain and source), increasing as hydrogen ion concentration in solution increases. Silicon dioxide and silicon nitride are two common choices for the insulator layer in an ISFET and are particularly attractive for low-cost pH sensors because they are compatible with established integrated circuit fabrication processes and as a result, are low-cost and easily mass-produced. 

Despite initially emerging in the scientific literature in the 1970s, the ISFET was not commercialized until the 1990s because irreversible reactions with the insulator layer make the devices vulnerable to drift. Sensitivity to both light and temperature further compromises ISFET performance. Further, like the glass electrode, the ISFET requires a reference electrode which is both fragile and difficult to miniaturize. In spite of these limitations, some pH ISFETs are commercially available (e.g., through Emerson, ThermOrion, Honeywell, and Sentron) and are suitable for monitoring fruit quality at various stages from the point of harvest through post-harvest processing and transport. However, the cost of these ISFET devices remains relatively high and limits their practical use in farm-to-table applications. 

The reference FET (REFET) provides an alternative to the traditional reference electrode-based electrochemical cell by using two ISFET structures. The first ISFET is pH sensitive and the second is not, thereby providing a reference point for pH measurements [74]. Unfortunately, REFETs demonstrated to date have been hampered by a short lifetime that prevents their commercialization. Another modification of the basic ISFET structure is the extended gate FET (EGFET) which separates the electrical and the chemical components of the ISFET system (Figure 5c) to preserve the lifetime of the electronics, promote stability, and limit drift [75]. For high resolution, dual-gate ISFETs [76] use two gates rather than one. Manipulation of the capacitance associated with one gate vs. that associated with the second gate on the dual-gate FET has successfully and dramatically increased the Nernstian limited sensitivity of 59mV/pH unit associated with other pH sensors. At this time, however, difficulties in commercializing these ISFET alternatives have made miniaturizing the traditional reference electrode design the most lucrative among the electrochemical alternatives to the glass electrode. MicroSens makes an integrated ISFET-based pH sensor using such a miniaturized electrode [77]. 

A wide range of electrochemical approaches to measuring pH are available at an equally wide range of costs and all with limitations. These approaches are summarized in Table 6. Note that a pH range between 2 and 7 is suitable for monitoring the acidity of fresh fruit. 

### 4.4. Electrical Approaches to pH Measurement

An alternative to electrochemical sensing of pH is the direct conversion of pH to electrical resistance or conductivity. This approach to sensing has the potential to be compact, low-cost, and low power. pH sensitive, conducting polymers include polyaniline or PANI [86] are sensitive to the pH ranges associated with fruit quality. Instability in sensor output associated with these films when used as pH sensors can be addressed through the addition of other polymers such as polypyrrole [87]. Some metal oxides are also suitable for sensing pH. For example, ruthenium oxide deposited on an interdigitated silver electrode experiences a decrease in resistance with increasing pH in a range between 3 and 11 pH units [88]. Titanium monoxide is also sensitive to pH in solution, but poor repeatability makes this material suitable only for coarse or single use measurements [89]. At low frequencies, different types of tin oxide films also show a decrease in conductance with increasing pH in the range of 2–11 pH units [89]. While these conductance- or resistance-based measurements are noisy and likely to offer lower resolution than electrochemical sensors, they nevertheless offer a level of simplicity that could enable sufficiently low-cost manufacturing to support monitoring pH in individual fruit. If sufficiently small amounts of juice (or fruit mash) allowed for an adequate sample, non-destructive evaluation of pH in fruit would also be possible. 

### 4.5. Electromechanical Approaches to pH Measurement

pH can also be measured using electromechanical techniques which involve converting a change in pH to a change in one or more mechanical properties of the sensor such as bending, mass, shape, or elasticity. One of the most popular approaches to extract pH using electromechanical sensors is the micro or nanoscale cantilever beam. In this type of sensor, the beam is coated with a material that bends in response to changing pH in the ambient environment. The bending of the beam is then read using the capacitive, resonance, or piezoresistive properties of the accompanying sensor.

Hydrogels are an attractive coating for cantilever beams. They undergo significant conformal changes in shape and size in response to pH changes. A hydrogel is a polymer (plastic) that contains molecular chains which are cross linked into a 3D matrix. In the first phase of the hydrogel’s response to a solution, the hydrogel is at its most hydrophobic (i.e., repelled from water) and shrinks. In the second phase, the hydrogel tries to mix with the solution in which it is immersed, thus causing it to expand or swell at maximum hydrophilicity (i.e., attraction to water). Fortuitously, some hydrogels experience this transition at a point that is dependent on pH [90]. When bonded to a cantilever beam for signal detection, the large changes in shape of the hydrogel—up to 100 fold in response to small changes in pH, can compete with the sensitivity of electrochemical sensors. Hydrogel-based pH sensors using cantilever beam technology have demonstrated sensitivities on the order of 10^−5^ pH units [90]. However, this high sensitivity comes at a price because the limited motion of tiny cantilever beams limits the dynamic range of the resulting pH sensor. For broader dynamic range, albeit at lower sensitivity, other pH-sensitive materials such as silicon nitride and silicon oxide can be deposited on microcantilevers to detect pH [91]. Although other structures such as diaphragms are available to facilitate this type of sensing approach, the microcantilever beam is both compatible with conventional microfabrication processes and highly sensitive to small changes in the beam properties. While the use of a cantilever beam may seem unnecessarily complex for measuring pH, this approach has been successfully used in a wide range of accelerometer designs at low cost. Thus, arrays of hydrogel coated beams are an option for high-sensitivity pH sensors and beams coated with silicon nitride or silicon oxide offer a less-sensitive but low-cost solution that can capitalize on existing commercial MEMs (microelectromechanical systems) fabrication processes. 

Unfortunately, electromechanical systems can be limited by their ability to detect motion of the cantilever beam or other mechanical structure using piezoresistive or capacitive means. Alternatively, coupling the mechanical signal to an optical rather than electrical signal can offer much higher detection limits. Such nano-optomechanical systems can achieve motion detection limits on the order of femtometers and can enable single molecule detection in liquids [92]. While these optomechanical sensors offer the small size appropriate for sensing pH while fruit are in storage or in transport, their cost may not be compatible with the small volume applications that are typical of farm-to-table operations. Nevertheless, they offer a viable alternative to electromechanical means for measuring pH and future advances in sensor development may significantly reduce the costs associated with this approach. 

### 4.6. Optical Approaches to pH Measurement

In addition to measuring pH optically via the mechanical motion of a pH-sensitive element, pH can also be measured indirectly with other optical techniques. For example, a pH-sensitive dye can be immobilized in a solid material or matrix that is designed so that the protons or hydronium ions that are indicative of pH can be measured via RI (refractive index). Using RI as an indicator of pH has some advantages. While conventional pH color strips produce a pH-sensitive color that can be measured with the naked eye, the eye can be inconsistent, unreliable, and imprecise. More precise measurements of color require a colorimeter to precisely measure the absorption of light by different colors or a spectrophotometer to measure the transmittance or reflectance of light associated with a color strip. This usually makes the resulting instruments expensive. In contrast, RI can be measured electronically and precisely at much lower cost (Table 4) than color.

Another optical approach to detecting pH involves using materials which are both pH sensitive and which exhibit fluorescence. Fluorescent pH dyes work by absorbing light across a relatively wide range of colors or wavelengths and re-emitting that light at a different and distinct color. Fluorescence intensity typically changes with pH, although some fluorescent dyes change in phase or color in response to pH. pH-sensitive, fluorescent dyes offer higher resolution than many other techniques and operate over ranges compatible with fruit [93], but they contaminate samples during measurement which eliminates any possibility of non-destructive evaluation of intact fruit. 

Also complicating optical pH sensing approaches is the fact that light signals must be subsequently converted to an electrical signal. Adding an accurate photodiode or other photodetector to these systems adds overhead, power consumption, and cost to the overall instrument. Despite the increased overhead, optical pH detection is more stable and less vulnerable to interference than other techniques. 

## 5. Sensing Technologies for Ethylene 

The presence of ethylene gas in the ambient environment can influence fruit ripening at levels as low as tens of nL/L [94]. During the ripening of a climacteric fruit such as the tomato, ethylene production can increase as much as 20 fold before starting to decline as the fruit over-ripens and degrades. Daily changes in internal ethylene concentration of tomatoes from ripening fruit are on the order of 100 nL/L [95]. Determining the optimal ripening point for harvest in farm-to-table operations depends on how far the fruit must travel to reach the point of consumption as well as the conditions under which it will be processed and transported. If these things are known, measuring the rate of change of ethylene concentration and the direction of change can play a critical role in optimizing the time of harvest and maximizing yield. Unlike conventional agriculture, ambient ethylene concentration is not artificially manipulated in farm-to-table operations because the delay between harvest and consumption is much shorter. Thus, fruits can be harvested when they are almost at peak ripeness rather than well before they have fully matured. This allows for fresher, more nutrient rich, and higher quality fruit to reach the consumer. Further, while ethylene sensors are useful at the point of harvest to monitor the rate of change of ethylene production and accurately identify the ripening stage of fruit, they are also useful post-harvest. After a fruit has been picked, it is likely to be boxed with companion fruit and produce of other types in shipments direct to the consumer. This can trigger ethylene-sensitive fruits (Table 3) to begin ripening faster than normal, risking premature spoilage. Therefore, post-harvest, ethylene sensing may be reduced to threshold sensors which trigger an alarm only when one or more thresholds of ethylene sensitivity have been reached. These thresholds depend on the type of fruit or vegetables shipped in close proximity to one another. Threshold-based ethylene sensors that are triggered at higher concentrations open up possibilities for low-cost, single-use, and low or zero-power sensors to support fruit quality throughout post-harvest processing. 

Four sensing technologies presently dominate ethylene gas sensing in agriculture and food distribution: photoacoustic spectroscopy, gas chromatography, non-dispersive infrared spectroscopy (NDIR), and electrochemical sensing. Each approach offers limits of detection in the parts per billion (ppb) range but with unique pros and cons that limit their use in farm-to-table operations. These four primary sensing technologies are discussed next, followed by a look at other chemical sensors that may offer lower-cost options. 

### 5.1. Photoacoustic Spectroscopy 

Photoacoustic spectroscopy uses a microphone in a controlled testing chamber to measure pressure changes that follow temperature shifts incurred when infrared light from a laser is absorbed by a gas of interest inside the chamber. For ethylene, a CO_2_ laser with emission at 10,600 nm is an excellent match to ethylene gas. Ethylene actively absorbs IR light around this wavelength [96]. The laser light source is both polarized (vibrating in a single plane) and chopped (turned on and off) so that the photoacoustic signal can be measured with high precision. Under controlled laboratory conditions using gas lasers, ethylene gas concentrations as low as 6 parts per trillion (ppt) have been measured [97]. Field measurements of ethylene gas using photoacoustic spectroscopy have not surprisingly demonstrated limits of detection that are not as good as laboratory results. Nevertheless, detection limits compatible with fruit quality monitoring are possible. In the field, photoacoustic spectroscopy using gas lasers has detected ethylene gas between 18.7 and 40.3 ppb [98] and between 0.6 ppm and 47 ppm [99] when applied to monitoring air pollution. Using a smaller, less power-hungry (0.3 mW) semiconductor laser, ethylene has been detected at 30 ppm [100]. Further, using quartz enhanced photoacoustic spectroscopy in combination with low power semiconductor lasers has allowed limits of detection as low as 300 ppb [101]. Advances in the use of semiconductor lasers for photoacoustic spectroscopy have led to the commercialization of mobile ethylene detectors with detection limits in the hundreds of ppb. However, these instruments are expensive (tens of thousands of dollars), power hungry (100 W), and heavy (13 kg) [102]. Even less-sensitive photoacoustic spectroscopy instruments suitable for detecting levels of ethylene in farm-to-table operations are far too expensive and bulky. Photoacoustic sensors are also inherently sensitive to noise and vibration in the ambient environment as well as temperature and humidity changes in the gas under test [32]. These vulnerabilities further detract from their appeal as portable instruments for monitoring ethylene gas production and absorption by fruit. 

### 5.2. Gas Chromatography 

Gas chromatography is another popular and viable approach for detecting ethylene gas. While not a sensing method in and of itself, gas chromatography serves a key function in ethylene gas sensing by separating ethylene from other gases in a sample in order to reduce interference from other similar gases. Once gases are separated by the gas chromatograph, their concentration can be sensed using a variety of methods of which flame ionization detectors (FID), thermal conductivity detectors (TCD), electrochemical sensors, and mass spectroscopy are among the most common. A gas chromatograph uses a carrier gas which is either inert (e.g., helium) or unreactive (e.g., nitrogen) to transport a sample (of gas or vapor) into a column where it interacts with a liquid or polymer in the column. Different gases are eluted (released) from the column at different times so that the detector at the back end of a gas chromatographic system must only sense one gas at any given time. In combination, the time after introduction of the gas to the chromatograph and the sensor (or detector) output determine the type and concentration of the gas, respectively. Automated samplers available for commercialized systems also offer a level of repeatability in measurements that is not possible with manual introduction of sample gases to gas chromatography systems [94]. 

Multiple detector technologies are available to support gas chromatography. Flame ionization detection (FID) works by mixing the carrier gas (and eluted gases) from the gas chromatography column with hydrogen and burning them in a flame. Approximately one in 10,000 molecules from the gas of interest in the burning process creates a gas-phase ion which is collected by an electrode that is placed above the flame. The resulting current is highly sensitive and proportional to the concentration of ethylene or other gas of interest. The thermal conductivity detector (TCD) works by monitoring the temperature of a hot filament as the carrier gas and eluted gas from the gas chromatograph pass by the filament. Since the carrier gas and eluted gas have different thermal conductivities, the temperature of the filament will change depending on the proportion of eluted gas in the sample [103]. Temperature changes can be sensed with high precision, resulting in a highly sensitive gas sensing mechanism. When mass spectroscopy is used as the detector in a gas chromatography system, the sample (carrier gas + eluted gas) is ionized by bombarding it with electrons and then separated according to mass-to-charge ratio of the two gases by accelerating them and deflecting them through an electric or magnetic field. Charged particles with the same mass to charge ratio will be deflected similarly and can be collected by a device such as an electron multiplier. Since the amount of deflection is related to the mass of the gas, the mass spectrometer provides further selectivity in ensuring at any given time and at any given detector in the sensing system, only one gas is sensed at a time [103]. In summary, gas chromatographs paired with any of these three forms of detection can be highly sensitive and highly selective to ethylene. However, these systems are also large and complex, consume significant power, and are very expensive. Thus, while traditional gas chromatography systems are well suited for precise analysis and modelling of fruit ripening, they are poorly suited to the cost and power constraints faced by small farms and small businesses involved in farm-to-table operations. 

The miniaturization of gas chromatography systems, however, offers some hope for their use in monitoring fruit ripening and fruit quality. Miniaturization of the column used in gas chromatography inherently reduces the sensitivity of the column as less surface area is available to capture and release gases of interest. To improve sensitivity in these miniaturized systems, a pre-concentrator amplifies the concentration of gases of interest to enhance the overall sensitivity of the gas chromatography system [104]. Pre-concentration can improve the limit of detection of ethylene detection systems from 140 ppm to 6 ppm [105] and 1 ppm [106]. However, preconcentration also introduces humidity, which impairs the accuracy and limit of detection for these systems. Despite these limitations, recent advances in the use of micromachining to construct three dimensional GC columns have demonstrated limits of detection for ethylene gas in banana monitoring of 35 ppb [107] and a resolution of 12 ppb using metal oxide detectors [108]. These performance levels are compatible with monitoring both ethylene gas concentrations and rates of change in these concentrations relevant to monitoring fruit ripening and degradation. However, even miniaturized systems are expensive and cost prohibitive for many small farms and businesses. But, the ultra-sensitivity of nano-optomechanical systems at the detector (back-end) of GC systems has the potential to replace preconcentration while still providing ppm level sensitivity for volatile organic compounds and also enabling miniaturized and low-cost systems for ethylene sensing [109]. 

### 5.3. Non-Dispersive Infrared Spectroscopy (NDIR)

In contrast to gas chromatography and photoacoustic spectroscopy, non-dispersive infrared (NDIR) spectroscopy can analyze and detect gases with far fewer stages of detection and lower overall complexity. NDIR offers great promise for portable and low-cost sensing applications. Infrared spectroscopy is already a familiar part of agricultural operations and has been demonstrated for a wide range of uses including detecting adulterants in beef products [110] and wine, olive oil, and fruit juice [111], pesticide residues in strawberries [112], moisture in grains, protein, oil, and soybeans, and quality of spices, teas, medicinal plants, fruits, vegetables and dairy products [113]. NDIR sensors are some of the simplest spectroscopic sensors because they do not require specialized optics such as prisms or diffraction gratings to disperse or separate incoming light to the sensor. 

NDIR sensors operate in one of two ways. In one configuration, an infrared light source is paired with an optical filter to select a narrow band of incoming IR light. The light is then transmitted through a gas sample in a light tube to an infrared photodetector. The light source and optical filter are selected to transmit light that is absorbed by the gas of interest. Incoming light is also chopped (turned on and off) or otherwise modulated to reduce interference from heating effects caused by the light source itself. Ethylene absorbs infrared light well at 10,600 nm, which makes it compatible with this form of NDIR sensing. However, ethylene is not the only gas that absorbs infrared light at this wavelength and interference from carbon dioxide and other gases commonly found in air can compromise the accuracy of this type of NDIR configuration. A second NDIR configuration uses multiple infrared detectors each of which evaluates IR light absorption at a different band of wavelengths in order to reduce the effect of interference. NDIR sensor systems of this type have been successfully demonstrated to detect ethylene at a limit of detection of 5 ppm [32]. Commercially available systems using this approach to NDIR sensing detect ethylene at limits of detection on the order of 20 ppm [114]. Pre-concentrators are also an option to improve the limit of detection and have demonstrated 40-fold improvements in performance [115]. NDIR systems have been successfully demonstrated in monitoring the artificial ripening of fruit with 5 ppm detection limits [116], thus underscoring their suitability in both performance and footprint to detecting ethylene for monitoring fruit quality. While NDIR limits of detection and resolution have room for improvement, the simplicity, low cost, and inherent stability of this optical sensing technique is one of the most promising opportunities for monitoring ethylene in farm-to-table operations.

### 5.4. Electrochemical Sensors

Traditional electrochemical sensors produce one of two types of electrical outputs: a current or a voltage. When the output of the sensor is a voltage, the sensor typically draws no current and its operation is labelled potentiometric. The voltage across the potentiometric electrochemical sensor is linear and a Nernstian function of the concentration of the analyte of interest (e.g., the pH sensor of Equation (4)). For example, an ethylene electrochemical sensor made with a Fe_0.7_Cr_1.3_O_3_ working electrode and a solid electrolyte has been demonstrated for measuring ethylene at high temperatures in diesel exhaust [117]. Ethylene was detected at levels down to 50 ppm for this potentiometric sensor with a Nernstian sensitivity of 0.12 mV/ppm. A level of 50 ppm is too high a detection limit for monitoring fruit quality and potential degradation. Unfortunately, these relatively high detection limits are only one problem associated with a potentiometric approach to measuring ethylene. Potentiometric sensors have limited selectivity. For example, the Fe_0.7_Cr_1.3_O_3_-based diesel exhaust sensor demonstrated in [117] was also sensitive to carbon monoxide (CO) at 0.024 mV/ppm. CO is present in high concentrations in diesel exhaust thus making a 5:1 selectivity ratio impractical for sensing ethylene in these applications. Similar problems in selectivity are encountered in monitoring fruit quality because of interference from ethanol and other fruit metabolites. Further, humidity can have a dramatic effect on the performance of electrochemical gas sensors in potentiometric operations, producing 20% differences in sensor sensitivity between dry and wet conditions [117]. Thus, despite the stability and predictability of potentiometric electrochemical sensors, their vulnerability to interference from related gases and from humidity often compromises their candidacy for measuring ethylene in agricultural applications.

Unlike potentiometric operation, the amperometric operation of electrochemical sensors produces a current that reflects the concentration of an analyte of interest in the sensing environment. This current flows from counter to working electrode in the electrochemical cell or vice versa. Drawing current between the electrodes in an electrochemical sensor improves sensitivity, lowers the limit of detection, and overcomes selectivity issues associated with potentiometric sensors. Amperometric electrochemical sensors for detecting ethylene gas have been implemented using ionic liquid as the electrolyte between electrodes for a limit of detection of 760 ppb [118] and using a compact, three-electrode measurement system that includes a micropump for circulating ambient air over the sensing electrode to sense ethylene down to 100 ppb [119]. While these limits of detection are attractive for ethylene monitoring during fruit ripening as well as during post-harvest processing and transport, this approach has significant drawbacks. For ethylene, acid electrolytes are often used in conjunction with gold working electrodes to prevent gold oxide from forming before ethylene can oxidize the electrode; acid toxicity and corrosion are undesirable for portable instruments and limit sensor and instrument lifetime. Further, when ethylene oxidizes a working electrode, it produces interfering gases which can compromise the overall selectivity of the sensor and further shorten its lifetime [94]. Amperometric operation also consumes electrode material and electrolyte which leads to drift and stability problems in sensor performance. Thus, in practical terms, stability, degradation, and lifetime issues in amperometric sensors and the poor selectivity of potentiometric sensors must still be addressed in order to make electrochemical sensors viable alternatives for sensing ambient and internal ethylene in fruit. 

### 5.5. Other Chemical Sensors

Electrochemical sensors are distinctive from other sensors which convert chemical information directly to electrical information in that they involve oxidation and reduction reactions with conductive electrodes to generate a current or voltage indicative of chemical concentration. Electrochemical sensor systems require two and sometimes three electrodes to ensure stable, drift-resistant operation. These requirements increase the size and cost of the overall instrument. Other means for directly converting chemical information to an electrical parameter are possible. For example, a *chemiresistor* converts chemical information directly to a change in conductance or resistance via reactions with a gas of interest in the ambient environment. Metal-oxide semiconductors are particularly attractive for use as chemiresistors because their baseline resistance is low compared to other materials and they are sensitive to a wide range of reducing gases. In a metal-oxide chemiresistor (Figure 6), reducing gases, including ethylene, interact with oxygen on the surface of the semiconducting metal oxide, thereby causing electrons to be re-injected into the bulk semiconductor and increasing conductivity (decreasing resistance).

Tin oxide (SnO_2_) is a common n-type semiconductor used for many decades in gas sensing. Tungsten oxides [120] and iron oxides [121] are also sensitive to ethylene gas but without modification exhibit lower sensitivity and less favorable detection limits than tin oxide-based sensors. Regardless of the type of metal oxide used in these gas sensors, the circuits and interfaces required to measure subsequent changes in resistance are simple and offer the possibility for low-cost, compact, and perhaps even single-use ethylene sensors. However, unmodified metal oxide semiconductors are limited in detection limit and sensitivity by their surface-to-volume ratio, are broadly selective to a range of reducing gases, and drift over time as a result of irreversible reactions with the exposed surface. Significant research attention has been directed at improving these three performance limitations. For example, the addition of catalysts to metal-oxide gas sensors increases selectivity to certain gases over others. For example, adding palladium (Pd) nanoparticles as a catalyst to tin oxide-based gas sensors has been shown to increase the ethylene response of these sensors by a factor of 3X while providing detection limits as low as 50 ppb [122]. The use of heterostructures can also improve sensor performance. For example, cerium oxide-tin oxide nanocomposite heterostructures can increase sensor response to ethylene gas by a factor of 5 or more and reduce the detection limit from ppm to sub-ppm levels [120]. Further, nanoparticles and nanostructures offer increased surface area as a percentage of overall sensor volume for increased sensitivity and lower detection limits. Tin oxide nanoparticle sensors enhanced with palladium respond to ethylene concentrations in the tens of ppm [123]. Carbon nanotubes offer even higher surface-to-volume ratios than nanoparticle-based gas sensors by facilitating the adsorption of gas molecules (including ethylene) on both the interior and exterior of their hollow tubelike structures. Copper ligands have been mixed with single walled nanotubes (SWNTs) to fabricate highly stable ethylene sensors with sub-ppm detection limits and a selective response to ethylene over other fruit metabolites—a 3-fold response compared to acetaldehyde and 8-fold response compared to ethanol [124]. Other carbon nanotube-based gas sensors are based on Wacker oxidation that does not require the binding of copper to ethylene but instead uses a palladium catalyst to enhance the oxidation process and increase the conductivity of the nanotube. This approach to ethylene sensing also offers sub-ppm detection limits as well as response times on the order of seconds [125]. Advanced structures such as graphene oxide-modified iron oxides have pushed detection limits down to 10 ppb while still retaining response times on the order of seconds [121]. These fast response times along with the simplicity of many chemiresistors continue to make them attractive for small and low-cost ethylene sensors.

Other materials have also demonstrated some potential for chemiresistive-based ethylene gas sensing. The perovskite LaFeO_3_ responds to ethylene at 150 °C but interference from acetylene gas and detection limits in the hundreds of ppm [126] limit its usefulness in farm-to-table operations. Still other structures and materials offer improvements in selectivity that are critical to the commercial use of these sensors in farm-to-table and other farming operations. The porous structure of metal organic frameworks (MOFs) offer ample opportunity to make size and chemical selective gateways for a variety of gas sensing materials. MOFs based on tetrahedrally-coordinated transition metal ions such as iron, copper, and zinc and connected by imidazolate linkers (e.g., ZIF-8) are especially attractive for low-cost ethylene sensors as they have demonstrated excellent ethylene adsorption properties [127]. 

The sensitivity of metal oxides to reducing gases such as ethylene can also be exploited using capacitance as the electrical output parameter. Tin oxide nanoparticles sandwiched between two copper electrodes have been demonstrated as effective *chemicapacitors* for detecting ethylene down to ppm ranges [128]. Selectivity to ethylene over other reducing gases can be enhanced by adding palladium or platinum between the electrodes and the metal oxide, but at the expense of faster degradation and shorter sensor lifetime [128]. Measuring capacitance rather than resistance offers similar short response times to resistance-based sensors while enabling ultra-low-power operation made possible by reading a signal from an insulator (via a capacitor) as opposed to a conductor (as is the case with a chemiresistor). 

## 6. Opportunities for Chemical Sensors in Farm-to-Table Operations 

While those who purchase and consume their food within farm-to-table operations often desire to support the local economy, sustain small- to medium-size family farms, and reduce the carbon footprint of what they eat, these are often not the primary goals of choosing farm-to-table over conventional corporate agriculture. Rather, the vast majority of those who access their food supply through farm-to-table pathways, whether home cook, restaurants, or entire chains of restaurants, are focused on the improvements in quality and freshness that farm-to-table provides. In many ways, the defining characteristics of farm-to-table operations naturally improve quality. For instance, sourcing locally and directly from the farm reduces the time between harvest and consumption, thereby improving the nutritional value of both fruits and vegetables. Vastly reduced transport, distribution, and storage times also enable picking produce at near-optimal ripening rather than picking immature crops and artificially enhancing ripening (or delaying ripening) along the long road to the consumer. More optimal harvesting times, in turn, improve the balance between sweetness and acidity that ultimately determines the flavor of fruit at the point of consumption. If the only objective of farm-to-table were to deliver the same flavor of fruits and vegetables that corporate agricultural operations offer, additional sensing technology would be largely unneeded. However, as expectations for better flavor, better texture, and better overall fruit quality continue to rise, and particularly so for farm-to-table operations, integrating sensors along the pathway from pre-harvest to consumption becomes more and more essential to keeping up with consumer expectations. 

While there is far more to flavor, taste, and overall perceptions of fruit quality than sweetness (measured as sugar content), acidity (measured as pH), and ethylene (indicative of ripening stage), these three parameters provide a foundational chemical baseline for establishing and tracking fruit quality. When it comes to measuring sweetness in terms of total sugar content, most sensing technologies offer resolutions on the order of one-tenth or one-quarter of one degree Brix (Table 4) which is more than sufficient to monitor changes in the sugar content of fruit from poor to excellent according to conventional °Brix charts (Table 1). Optical methods (e.g., near-infrared spectroscopy) offer the added advantage of non-contact and non-destructive measurements at prediction errors of less than a single degree Brix (Table 5). Portable NIR spectrometers are available but their costs still run between $300 and $5000 USD [70]. This is substantially more than low-cost refractometers and hydrometers which offer similar resolution and accuracy in estimating total soluble solids content and total sugar content. A compromise between non-destructive but expensive and destructive but inexpensive approaches to measuring fruit sweetness may lie in the sampling approach. Advances in microsampling and microextraction of the internal solids of fruits have the potential to exploit the low-cost and low-power operation of refractometers and hydrometers while not visibly damaging the fruit, compromising its ripening behavior, or accelerating its degradation. While using RI or specific gravity as a measure of sweetness remains prone to errors from non-sucrose sugars and from non-sugar soluble solids, differentiating among these different forms of soluble solids is likely not necessary for estimating flavor and monitoring changes in sugar over time as fruit travels from harvest to consumption. 

Estimates of flavor based on total sugar content alone can be highly inaccurate because acidity plays also plays an important role in perceived sweetness and overall flavor. Although titratable acidity offers a more accurate representation of total hydrogen ions and acid content in fruits, measuring it requires crushing and filtering the fruit as well as adding a known base to the resulting juice until a certain pH is reached. This method is inherently destructive and as a result, pH (i.e., free hydrogen ions in solution) is frequently used as a proxy for titratable acidity in evaluating fruit and fruit juice. Unlike sensors which determine sugar content to varying degrees across a wide range of price points, there is a large gap between pH sensors of coarse resolution (on the order of 1 or 0.1 pH units) and higher performance pH meters which rely on electrochemical means to determine pH. Like sugar sensors, however, pH sensors require converting intact fruit to mash which is inherently destructive to the fruit. The lowest cost alternatives to pH sensing involving strips that use multiple litmus indicators are vulnerable to human errors in judging the color of the strip, especially when stained by the fruit itself. Low-cost pH sensors (less than $40) based on electrochemical sensing techniques are commercially available (Table 6) but calibration packages for these sensors can double the cost and a significant volume of liquid sample is still necessary. Viable, non-destructive techniques for pH sensing in farm-to-table operations may, like sugar content, involve using microextraction techniques to sample a small, low-impact volume of juice from the fruit. These low volume samples might then be analyzed using a custom calibrated litmus-indicator under controlled lighting conditions that offers resolution and accuracy on the order of tenths of pH units. Equally as viable are miniaturized electrochemical pH sensors which function with sufficiently small sample sizes to avoid significant degradation of the fruit being tested. 

While sweetness and acidity contribute directly to perceptions of flavor and freshness, ethylene production among climacteric fruits and absorption by all fruits (and vegetables) plays a critical role in predicting flavor and freshness at the point of consumption. Too little ethylene in a climacteric fruit indicates a fruit not yet ready to be picked. Too much ethylene introduced on the route to the consumer poses a threat to all fruits in the vicinity of the excessive ethylene producer. Rates of change in ambient ethylene also provide valuable information about how much time a batch of fruit has before it becomes unsuitable for consumption. Unfortunately, most ethylene sensing technologies that are accurate and resistant to interference from other ambient gases are expensive and the performance of lower-cost alternatives is affected by ambient fruit metabolites such as ethanol. Nevertheless, chemiresistors and other low-cost ethylene sensors have the potential to be used for tracking aggregate changes in ambient ethylene gas as fruits travel to the consumer, thereby providing a low-resolution alarm that indicates conditions which could compromise entire batches of produce. 

## 7. Conclusions 

The benefits of farm-to-table farming to local economies, to small farms, to the health of the consumer, and to reductions in greenhouse gas emissions are many. In order to remain viable and to justify the additional cost and effort involved in navigating these farm-to-table operations, diversification of what farms offer must be balanced with consistent quality and freshness. Monitoring the chemistry of farm-to-table products as they travel from the point of harvest to the point of consumption is a small but important part of ensuring that the best flavor and optimal freshness is what the consumer ultimately experiences from these products. For fruit, flavor is a function of both sweetness and acidity—sugar content and pH (or titratable acidity). Freshness at the point of consumption can be estimated from ripeness at harvest, temperature, humidity, and other ambient conditions during storage and transport, and by emissions of and exposure to ethylene gas. In farm-to-table operations, transit and storage times are typically much shorter than in larger, corporatized agricultural operations. Thus, fruit can be picked closer to peak ripening and are more likely to arrive at the point of consumption with optimal flavor.

For non-climacteric fruit, sensing ethylene while the fruit is still on the vine (or tree) is not nearly as important as it is for climacteric fruit. For the latter, peak ethylene emissions and peak ripening go hand in hand. Handheld ethylene sensing instruments that can accurately track small changes in day-to-day ethylene emissions from climacteric fruit would allow harvesting decisions to be made with much greater precision than is presently possible. The most likely candidates for these handheld instruments are miniaturized gas chromatographs coupled with pre-concentrators that are immune to or that compensate for humidity in the ambient environment or coupled with detectors at the back-end that offer low enough detection limits that preconcentration is not needed. These instruments are likely to remain expensive but shared use among multiple farms in the same community can improve access to this important technology. Once fruit are harvested, monitoring ethylene exposure becomes important for both climacteric and non-climacteric fruit. However, performance requirements for monitoring ethylene exposures and emissions are more relaxed. Instruments with high resolution and a continuous measurement range can likely be replaced with sensors that trigger only when ambient ethylene reaches levels that are known to trigger impending and premature degradation in certain types of fruit (or vegetables). In these scenarios, less accurate, more drift-prone, but dramatically less expensive sensors, such as metal oxide-based chemiresistors may deliver adequate performance over short transit and storage intervals to maximize yield and maintain freshness. 

As for sugar and pH, both are important for all types of fruit, whether climacteric or non-climacteric. Fortunately, low-cost options for sensing sugar content have been available for many decades and adjusting these options to farm-to-table operations is likely to be more about sampling than about sensing. Sugar must be measured in liquid samples and sensors or instruments that require low volume samples and sampling techniques that can non-destructively extract these liquid samples from intact fruit would open up a level of insight into the flavor of individual fruit that is out of reach with current technology. Similarly, pH must also be measured in liquid samples and having the capacity to monitor pH of fruit as it ripens, is harvested, and travels to its final destination without sacrificing individual fruit will be invaluable to tracking flavor and freshness. Unlike instruments and sensors that detect °Brix (sugar content), however, pH sensing instruments remain relatively expensive and costs must be reduced to improve accessibility and value to farm-to-table operations. Electrochemical pH sensors remain the most viable for pH sensing of fruit over the short term. Among these sensors, the need for a stable reference electrode is a major obstacle and investment in alternative paradigms that tolerate some instability in the electrochemical reference point may open the door to lower cost and possibly even single-use products for measuring pH. 

In summary, some problems associated with detecting pH, sugar content, and ethylene in farm-to-table applications are constrained by what available sensing technologies can do. As a result, these problems are likely to be resolved (or alternatively, their limits of operation ultimately identified) as part of continued research on sensing materials and sensor designs in the broader research community. Other problems, however, require more attention to the constraints and unique needs of farm-to-table operations. These problems will benefit from engineering miniaturized versions of existing sensor technologies and integrating sampling techniques, sensors, interface circuits, and signal processing into user-friendly, application-sensitive instruments that are economically accessible to multiple stakeholders in farm-to-table operations. In short, there are ample opportunities for advances in sensors research and instrument engineering to better serve and support the farm-to-table community. 

## Figures and Tables

**Figure 1 sensors-21-01634-f001:**
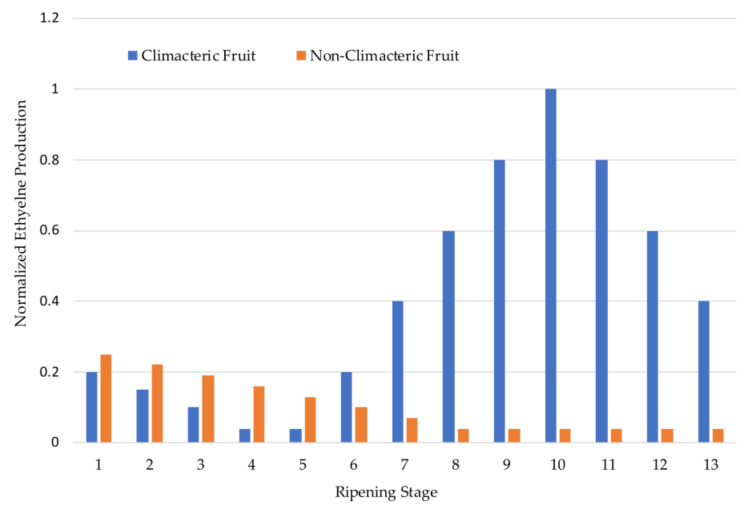
Ethylene Production in Climacteric and Non-Climacteric Fruits. Adapted from [29].

**Figure 2 sensors-21-01634-f002:**
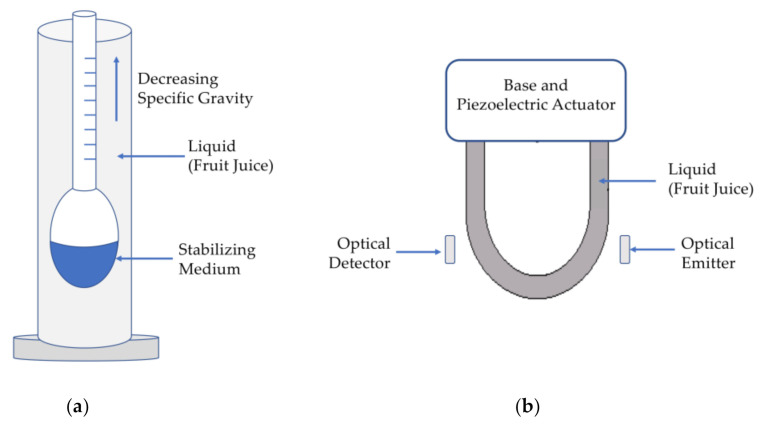
Instruments for Measuring Specific Gravity of Fruit Juice or Fruit Mash. (**a**) Typical hydrometer; and (**b**) oscillating U-shaped tube meter.

**Figure 3 sensors-21-01634-f003:**
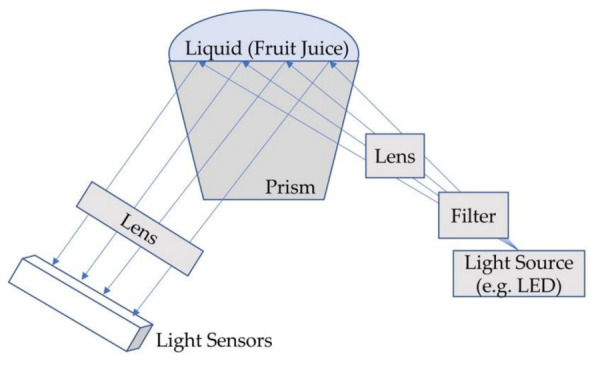
Traditional Refractometer Configuration.

**Figure 4 sensors-21-01634-f004:**
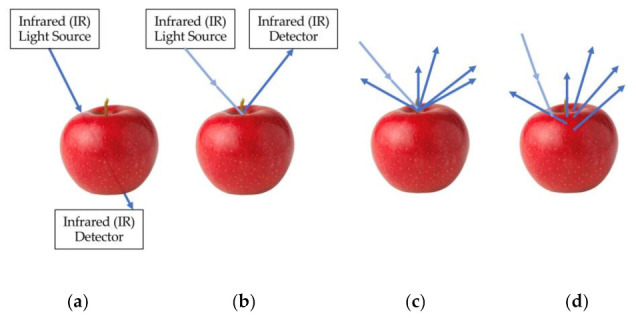
NIR Spectroscopy Configurations [59]. (**a**) transmittance; (**b**) reflectance; (**c**) diffuse reflectance; and (**d**) interactance.

**Figure 5 sensors-21-01634-f005:**
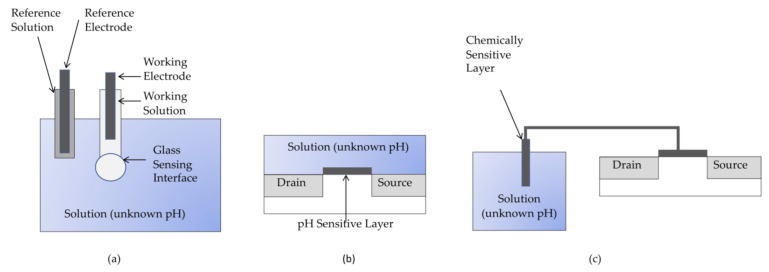
Electrochemical pH Sensors. (**a**) Traditional, glass electrode-based pH sensing; (**b**) ISFET-based pH sensing, where the gate of a field effect transistor (FET) is removed and is replaced by the interaction of solution and pH-sensitive layer to modulate the current between the drain and source of the transistor; and (**c**) extended gate FET (EGFET), which separates the electronic half of the ISFET from the electrochemically sensitive half in order to extend sensor lifetime and reduce drift.

**Figure 6 sensors-21-01634-f006:**
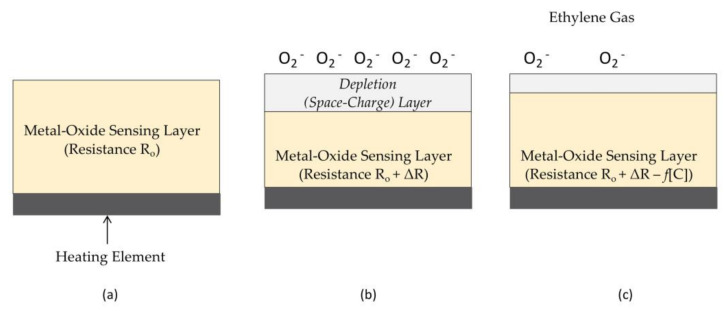
Metal-Oxide Gas Sensors. (**a**) In the absence of oxygen and reducing gases, the metal oxide has a baseline resistance on the order of kOhms. (**b**) Ambient oxygen on the surface of the sensing layer extracts oxygen from the underlying metal oxide, increasing resistance and creating a depletion (insulating) layer on the surface of the sensor. (**c**) A reducing gas such as ethylene binds with the oxygen on the surface, resulting in a re-injection of electrons into the sensing layer and a decrease in resistance that is a function of the ambient ethylene concentration. The heating element keeps the metal oxide sensing layer at an elevated temperature that maximizes the sensitivity of the sensor to desired gases.

**Table 1 sensors-21-01634-t001:** Sugar Content in Fruit across a Range of Fruit Quality [13].

Fruit	°Brix Associated with Quality of Fruit			
	Poor	Average	Good	Excellent
Apple	6	10	14	18
Banana	8	10	12	14
Cantaloupe	8	12	14	16
Cherries	6	8	14	16
Grapes	8	12	14	20
Grapefruit	6	10	14	18
Lemons	4	6	8	12
Mangoes	4	6	10	14
Oranges	6	10	16	20
Peaches	6	10	14	18
Pears	6	10	12	14
Strawberry	6	10	14	16
Tomato	4	6	8	12
Watermelon	4	6	8	12

**Table 2 sensors-21-01634-t002:** pH of Ripe Fruit [20].

Fruit	pH Range	Fruit	pH Range
Apples	3.10–3.40	Mangoes	3.40–4.80
Banana	4.50–5.20	Oranges	3.69–4.34
Cantaloupe	6.13–6.58	Peaches	3.30–4.05
Cherries	3.25–3.83	Pears	3.50–4.60
Grapes	2.80–3.84	Strawberry	3.00–3.90
Grapefruit	3.00–3.75	Tomato	4.30–4.90
Lemons	2.00–2.60	Watermelon	5.18–5.60

**Table 3 sensors-21-01634-t003:** Ethylene Production and Sensitivity in Fruit. NA: not available.

Fruit	Type [28]	Production [30,31]		Ethylene Sensitivity [32,33,29]
		Relative	μL/kg-h	
Apples	climacteric	high	10–100	high
Banana	climacteric	moderate	1–10	high
Cantaloupe	climacteric	NA	NA	very low
Cherries	non-climacteric	very low	<0.1	low
Grapes	non-climacteric	very low	<0.1	low
Grapefruit	non-climacteric	very low	<0.1	moderate
Lemons	non-climacteric	very low	<0.1	moderate
Mangoes	climacteric	moderate	1–10	high
Oranges	non-climacteric	very low	<0.1	moderate
Peaches	climacteric	high	10–100	high
Pears	climacteric	high	10–100	high
Strawberry	non-climacteric	very low	<0.1	low
Tomato, ripe	climacteric	moderate	1–10	low
Watermelon	non-climacteric	low	0.1–1.0	high

**Table 4 sensors-21-01634-t004:** Sensors for Measuring Sugar Content in Fruit and Fruit Juice (NS: not specified).

Instrument/Sensor	Ref	Accuracy	Range	Resolution	Cost
*Using specific gravity: SG Units (°Brix)*					
Hydrometer ^1^	[51]	NS	0.98–1.16 (0–40°)	NS	$12
Hydrometer ^1^	[52]	NS (0.50°)	NS (0–35°)	(0.5°)	$100
Hydrometer ^1^	[37]	NS	>1 (>0°)	1 × 10^−3^ (0.25°)	$46
U-Shaped Tube Meters ^2^	[53]	1 × 10^−3^ (0.25°)	0.0–3.0 (0–100°)	1 × 10^−4^ (0.025°)	$3290
*Using refractive index: RI Units (°Brix)*					
Refractometer (analog) ^1^	[54]	NS	NS	1 × 10^−4^ (0.1°)	$133
Refractometer (digital) ^1^	[55]	3 × 10^−4^ (0.20°)	NS	1 × 10^−4^ (0.1°)	$500
Refractometer (Abbe) ^2^	[56]	2 × 10^−4^ (0.14°)	1.3–1.7 (0–100°)	2 × 10^−4^ (0.14°)	$6630
Photonic Crystal Microcavities	[45]	NS	1.0003–1.0013 ^4^	1 × 10^−4^ (0.07°)	NA ^3^
Photonic Crystal Fibers w/long period gratings	[46]	NS	1.33–1.38 (0–33°)	8.5 × 10^−6^ (0.006°)	NA ^3^
Surface Plasmon Resonance	[41]	NS	NS	1 × 10^−7^ (0.000071°)	NA ^3^
Resonating Microspheres	[42]	NS	NS	1 × 10^−7^ (0.000071°)	NA ^3^
*Using capacitance: Farads (°Brix)*					
Parallel Plate Capacitors	[47]	NS	NS	NS (5.0°)	NA ^3^
Cylindrical Capacitors	[48]	NS	NS	4.22–5.02 pF (9.0°–10.9°)	NA ^3^
*Using the speed of sound: m/sec (°Brix)*					
Ultrasound (one sugar)	[49]	NS (0.2°)	(0°–40°)	NS	NA ^3^
Ultrasound (intact fruit)	[50]	NS	(1.5°–17.5°)	NS	NA ^3^

^1^ Available commercially as a portable instrument. ^2^ Available commercially as benchtop (laboratory) equipment. ^3^ Research; no commercial version available. ^4^ Measured for gases only; RI range is not matched to liquid sensing applications.

**Table 5 sensors-21-01634-t005:** NIR Spectroscopy for Measurement of Sugar Content in Fruit and Fruit Juice. NS: Not Studied; SEP: Standard Error of Prediction; RMSEP: Root Mean SEP.

Study	Approach	Type	Unit	R or R^2^	SEP or RMSEP
*Intact Fruits*					
[60]	reflectance	apples	°Brix	R^2^ = 0.94	SEP = 0.97
[61]	diffuse reflectance	apple, peeled	°Brix	R = 0.93–0.97	SEP = 0.37–0.42
[62]	transmittance	orange (mandarin)	SSC ^1^	R = 0.93	RMSEP = 0.65
[63]	reflectance	orange (mandarin)	SSC ^1^	R^2^ = 0.93	RMSEP = 0.32
[64]	reflectance	orange, navel	SSC ^1^	R = 0.90	RMSEP = 0.68
[65]	reflectance	grapefruit, red	SSC ^1^	R^2^ = 0.67	NS
[66]	diffuse reflectance	jujube	TS ^3^	R = 0.904	RMSEP = 0.26
[67]	interactance, transmittance	passion fruit	SSC ^1^	R = 0.92	NS
[68]	transmittance	pear	SSC ^1^	R^2^ = 0.87	RMSEP = 0.45
[69]	diffuse reflectance	strawberry	SSC ^1^	R^2^ = 0.94	RMSEP = 0.29
[70]	interactance	umbu fruit	SSC ^1^	R^2^ = 0.78	RMSEP = 0.72
	reflectance		SSC ^1^	R^2^ = 0.61	RMSEP = 1.02
*Fruit Juice*					
[71]	reflectance	orange	SSC ^1^	R = 0.98	RMSEP = 0.73

^1^ Soluble solids content used as an estimate for total sugar content. ^2^ R squared (coefficient of determination). ^3^ Total sugar content.

**Table 6 sensors-21-01634-t006:** Electrochemical Sensors for Measuring pH. NS: not specified; NC: not commercialized; NA: not applicable.

Technology	Sensitivity (mV/pH)	Resolution (pH Units)	Range (pH Units)	Cost (USD)	Ref
pH strips	NA	1	0–14	$0.02	
glass electrode	NA	0.001	−2–16	$400 ^1^	[78]
cloth junction	NA	0.1	0–14	$75 ^1^	[79]
ISFET	33	Varies ^2^	4–10	NC	[80]
ISFET with miniaturized reference electrode	55	Varies ^2^	2–11	NS	[81]
REFET	50	Varies ^2^	1–14	NC	[82]
EGFET	50	Varies ^2^	7–12	NC	[83]
EGFET	28	Varies ^2^	2–12	NC	[84]
dual-gate ISFET	304	Varies ^2^	3–10	NC	[85]

^1^ Includes calibration bundle. ^2^ Resolution depends on resolution of interface circuits or analog to digital converter.

## Data Availability

No original data generated by the author is contained in this article.

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
