# Peer review of "Chemical Sensors for Farm-to-Table Monitoring of Fruit Quality"

_sensors, 2021, doi:10.3390/s21051634_

Round 1

Reviewer 1 Report

Excellent review paper about monitoring of the quality of fruits.
The correlation of the RI coefficient with Brix is obvious since Brix is determined refractometrically.
The paper gives the excellent background for further development of applications of sensors for monitoring the quality of fruits during the short way transportation and distribution.

Author Response

Excellent review paper about monitoring of the quality of fruits.
The correlation of the RI coefficient with Brix is obvious since Brix is determined refractometrically. 

Response to Reviewer:

Thanks so much for this -- the figure has been replaced with a simple equation to express the relationship.  I agree that the Brix-RI relationship is obvious when the total dissolved solids in solution consist entirely of sucrose. But, when other sugars (e.g. fructose, glucose) are present or when other solids are present, this is not necessarily the case.  Clarifying notes have been made in the revised manuscript.  All significant changes to the manuscript are noted in blue.

The paper gives the excellent background for further development of applications of sensors for monitoring the quality of fruits during the short way transportation and distribution.

Response to Reviewer:

Many thanks for the time and effort given to reviewing the manuscript.  

Reviewer 2 Report

Reviewer: Minor revision

This review (sensors-1116994) focuses on the chemical sensors for monitoring sweetness (sugar content), acidity (pH) and ethylene gas. This topic is very interesting and meaningful, but there are still some problems to be solved before possible publication:

(1) Most of the references are out of date, so it is recommended to quote the latest references in this field.

(2) As a review article, it is suggested to quote representative diagrams (especially the sensors and their applications) to enhance the readability.

(3) Please check the format carefully. For example, indent the line 438; the row spacing of the 3.1, 3.4 and 4.6 sections. Title 3.4 repeated twice.

(4) “Conclusions” need to be strengthened. I would like to see more constructive strategies from authors.

(5) This review is insufficient in terms of the latest technologies and recent reports, please supplement appropriately. The following recent papers about ethylene/gas sensors may be used to expand your article: Chinese Chem. Lett. 2020, 31 (8), 2045−2049; ACS Appl. Mater. Interfaces 2020, 12, 28, 31037–31053; Sens. Actuators B Chem. 318 (2020) 128104.

Reviewer 3 Report

Insightful article. Recommend to publish.

My question to the author is insecticides are also used in fruit vegetation like Azoxystrobin, Difenoconazole, Captan, etc. Can the sensors be used in a farm to table concept to enquire whether those insecticides in the fruit are within the tolerance label?  

The author can also emphasize which commercially available sensors are right now and what the future market looks for sensors. I think the author can address these questions within this article, which might make it more informative.

Author Response

My question to the author is insecticides are also used in fruit vegetation like Azoxystrobin, Difenoconazole, Captan, etc. Can the sensors be used in a farm to table concept to enquire whether those insecticides in the fruit are within the tolerance label?  

Response to Reviewer:

Detecting insecticides at the low levels typical of farm-to-table and similar operations typically requires laboratory scale equipment (such as GC-MS) and is done using a batch or random sampling approach. These approaches were outside the scope of this article, but are certainly very relevant to organic and other alternative farming practices to elevate the value of these products to the consumer as well as other stakeholders.  

The author can also emphasize which commercially available sensors are right now and what the future market looks for sensors. I think the author can address these questions within this article, which might make it more informative.

Response to Reviewer:

Commercially available sensors are noted in various tables in the manuscript:  Table 4 and Table 6.  However, multiple comments have been inserted in blue in the revised manuscript to highlight and emphasis which sensing technologies are commercially available.  

Reviewer 4 Report

The author has presented a review article surveying the state of the art technology available for Non destructive (ND), non contact (NC) sensing of farm produce produced in a small scale operation. 

While I feel the author has given a bird's eye view of the available technology for ND, NC sensing, initially the author takes some time to describe technologies that have relevance in liquid sensing, which may not have much relevance in the farm to table context. I suggest the author to reduce the space alloted to those topics (section 3.1, 3.2, 3.3, 4.1 and 4.2) if possible, to keep the readers engaged.
Also I request some clarity in the author's description of corporatized agriculture. What is the yard stick to classify something as farm to table vs corporatized agriculture. Is it based on distance from farm to the serving community, revenue of the company, invested capital etc. 

I do have a general question about the premise of this article and this may be out of the scope of this article. While the premise of farm to table sounds interesting in terms of farm produce quality, I am not sure how far it is practical. In north american context, almost two-thirds of the land has cold conditions where agricultural activity has to be suspended for about 4 to 6 months in a year. In such case, corporatized agriculture might have the wherewithal to store and maintain the supply chain stability throughout the year. This seasonal weather changes creates a supply chain instability for farm to table firms (small operation as described by the author) and hence questions the viability of operation. While the quality of farm produce might suffer a bit in large scale operation, there is a tradeoff to be made here in terms of maintaining supply chain stability. And many of the sensing technologies that the author suggest are suitable for large operations as well. So how does the focus of this manuscript on sensing technologies for small scale operations alone justify. Can the author comment on that.

In addition I would like the author to include some previously published on different sensing technologies.
For section 4.5 on electromechanical measurement approaches to pH, I suggest the author to include work on Nanophotonic optomechanical sensors in the paper titled Improving mechanical sensor performance through larger damping, Science in which the authors describe nanophotonic sensors which has enhanced sensitivity under ambient conditions. this sensors may be suitable for non contact ambient storage sensing. This sensor is also compatible with hydrogel based surface coating for pH sensing.

For section 5.2 on gas chromatography, I suggest including the paper titled Nano-optomechanical systems for gas chromatography, Nanoletters, and Experimental Coupling of a MEMS Gas Chromatograph and a Mass Spectrometer for Organic Analysis in Space Environments, ACS Earth and Space Chemistry. In the former a lab demonstration and integration of the nanophotonic optomechanical sensor with a GC system is presented and in the latter the GC system is miniaturized for portable application suitable in scale for small operations.

For section 5.5, I suggest the author to include the paper titled Gas Adsorption characteristics of metal–organic frameworks via quartz crystal microbalance techniques, Journal of Physical Chemistry C, Direct synthesis of single-walled aminoaluminosilicate nanotubes with enhanced molecular adsorption selectivity, Nature communications and HKUST-1 coated piezoresistive microcantilever array for volatile organic compound sensing, Micro and Nano letters. These papers describe the potential of novel editable gas sensing materials like Metal Organic Frameworks, Alumino Silicate Nanotubes as well as their easy integration with simple, cost effective and portable Quartz crystal microbalance.  
